# π-π-Stacked Poly(ε-caprolactone)-*b*-poly(ethylene glycol) Micelles Loaded with a Photosensitizer for Photodynamic Therapy

**DOI:** 10.3390/pharmaceutics12040338

**Published:** 2020-04-09

**Authors:** Yanna Liu, Marcel H.A.M. Fens, Bo Lou, Nicky C.H. van Kronenburg, Roel F.M. Maas-Bakker, Robbert J. Kok, Sabrina Oliveira, Wim E. Hennink, Cornelus F. van Nostrum

**Affiliations:** 1Department of Pharmaceutics, Utrecht Institute for Pharmaceutical Sciences, Utrecht University, Universiteitsweg 99, 3508 TB Utrecht, The Netherlands; y.liu7@uu.nl (Y.L.); B.Lou@uu.nl (B.L.); n.c.h.vankronenburg@uu.nl (N.C.H.v.K.); R.J.Kok@uu.nl (R.J.K.); S.Oliveira@uu.nl (S.O.); W.E.Hennink@uu.nl (W.E.H.); 2Division of Cell Biology, Neurobiology and Biophysics, Department of Biology, Utrecht University, Padualaan 8, 3584 CH Utrecht, The Netherlands

**Keywords:** polymer micelles, photodynamic therapy, in vitro release, circulation kinetics, biodistribution

## Abstract

To improve the in vivo stability of poly(ε-caprolactone)-*b*-poly(ethylene glycol) (PCL-PEG)-based micelles and cargo retention by π-π stacking interactions, pendant aromatic rings were introduced by copolymerization of ε-caprolactone with benzyl 5-methyl-2-oxo-1,3-dioxane-5-carboxylate (TMC-Bz). It was shown that the incorporation of aromatic rings yielded smaller micelles (18–30 nm) with better colloidal stability in PBS than micelles without aromatic groups. The circulation time of i.v. injected micelles containing multiple pendant aromatic groups was longer (t½-α: ~0.7 h; t½-β: 2.9 h) than that of micelles with a single terminal aromatic group (t½ < 0.3 h). In addition, the in vitro partitioning of the encapsulated photosensitizer (meta-tetra(hydroxyphenyl)chlorin, mTHPC) between micelles and human plasma was favored towards micelles for those that contained the pendant aromatic groups. However, this was not sufficient to fully retain mTHPC in the micelles in vivo, as indicated by similar biodistribution patterns of micellar mTHPC compared to free mTHPC, and unequal biodistribution patterns of mTHPC and the host micelles. Our study points out that more detailed in vitro methods are necessary to more reliably predict in vivo outcomes. Furthermore, additional measures beyond π-π stacking are needed to stably incorporate mTHPC in micelles in order to benefit from the use of micelles as targeted delivery systems.

## 1. Introduction

Photodynamic therapy (PDT) is a minimally invasive form of therapy that has been approved for treatment of various types of cancers, including head and neck tumors, basal cell carcinoma, cervical, endobronchial, esophageal, bladder, and gastric cancers [1,2]. Compared to conventional treatments, such as surgery, radiation and chemotherapy, PDT has important advantages, including reduced toxicity for healthy tissues, lack of drug resistance mechanisms, and favorable cosmetic outcomes. Furthermore, evidence of immune activation has been described after PDT, capable of leading to antitumor immunity [3,4]. PDT is based on a photochemical reaction between a light activatable molecule (i.e., photosensitizers (PS)), light, and molecular oxygen, which are harmless individually [1]. In combination, PS can be excited by light of the wavelength matching its absorption maximum and can subsequently transfer its energy to molecular oxygen to yield singlet oxygen species (ROS), which are highly reactive with biomolecules present in cytoplasma or cell membranes, leading to cell death [5,6]. Effective cancer PDT is, however, hindered by some undesired properties in PS. Most PS molecules—including the clinically approved second generation PS, meta-tetra(hydroxyphenyl)chlorin (mTHPC)—are highly hydrophobic due to the extended delocalized aromatic π electron system. This promotes non-specific binding to cells, resulting in unspecific distribution of PS in healthy tissues (i.e., no selective accumulation of the PS in tumorous tissues), which can cause skin toxicity [7,8,9]. Moreover, the poor water solubility caused by the high hydrophobicity makes PS prone to aggregation in aqueous solutions, leading to lower ROS generation and decreased therapeutic efficacy [9,10,11]. 

Loading of PS in polymeric micelles is a promising approach to address these challenges [12]. Polymeric micelles are self-assembled nanostructures based on amphiphilic block copolymers formed in an aqueous solution and have been extensively investigated as drug delivery systems, particularly for the targeted delivery of hydrophobic drugs [13,14,15,16]. Polymeric micelles are characterized by a well-defined structure, containing a hydrophobic core and a hydrophilic shell. The micellar core has a high capacity to accommodate hydrophobic compounds, including photosensitizers, while the hydrophilic shell, which is mostly composed of poly(ethylene glycol) (PEG), can result in prolonged retention of polymeric micelles in the blood circulation by delaying their recognition and rapid uptake by the reticuloendothelial system (RES) [17,18,19,20]. In addition, micelles have customizable sizes ranging from 10 to 100 nm, which is favorable for the passively targeted delivery of loaded hydrophobic drugs to the aimed sites via the enhanced permeability retention (EPR) effect [21,22]. Previous research by our group showed that micelles based on poly(ε-caprolactone)-*b*-methoxy poly(ethylene glycol) (PCL-PEG) block copolymers can be loaded with the photosensitizer mTHPC with very high loading capacity [23]. However, in a recent study, we showed a rapid release of this PS in the circulation after i.v. administration of the mTHPC-loaded PCL-PEG-based micelles [24]. The poor stability in the circulation is currently considered a limitation for micelles for successful clinical applications [24,25,26,27]. This instability is most likely due to a combination of extraction of the cargo from the micelles and micellar destabilization resulting from a large dilution volume upon injection or binding of drug–polymer chains to blood components (e.g., albumin, lipoproteins) [28]. 

To overcome their inherent instability, physical interactions through π-π stacking have been investigated to enhance the stability of polymeric micelles. For example, Kataoka et al. previously reported that micelles consisting of poly(ethylene glycol)-poly(aspartate) block copolymers derived with multiple pendant 4-phenyl-1-butanol showed high paclitaxel retention in vivo [29]. Yang et al. demonstrated that paclitaxel-loaded micelles based on methoxy-poly(ethylene glycol)-*b*-poly(*N*-(2-benzoyloxypropyl)methacrylamide) (mPEG-*b*-p(HPMAm-Bz) had a significantly prolonged circulation time, good drug retention, and enhanced tumor accumulation, which resulted in substantially improved antitumor efficacy that was attributed to noncovalent stacking interactions between Bz groups and aromatic groups present in the drug [30,31,32]. In another study, thermosensitive HPMAm-lactate-based micelles that contained HPMAm-Bz (~30 mol%) units as comonomers were used to encapsulate a hydrophobic PS (i.e., Si(sol)_2_Pc, an axially solketal-substituted silicon phthalocyanine), showing enhanced loading capacity and significantly improved retention of PS during 9 days of storage in phosphate-buffered saline (PBS) at 37 °C as compared to Si(sol)_2_Pc loaded in HPMAm-lactate-based micelles lacking the aromatic comonomers [33]. 

The abovementioned studies prompted us to stabilize PCL-PEG micelles in the current study by π-π stacking interactions through introduction of multiple pendant aromatic moieties in the hydrophobic core. To this end, aromatic rings were incorporated in the hydrophobic polymer chains by ring-opening copolymerization of ε-caprolactone (CL) with a benzyl-functionalized trimethylene carbonate, namely benzyl 5-methyl-2-oxo-1,3-dioxane-5-carboxylate (TMC-Bz). For comparison, PCL-PEG diblock copolymers without and with one aromatic unit at the terminal PCL chain end were synthesized. Then, mTHPC was selected as a model PS with high aromaticity and hydrophobicity, and was encapsulated in polymeric micelles that were prepared from the resulting block copolymers. The effect of aromatic π-π stacking interaction on the loading capacity, possible PS aggregation in the micellar core, and stability of mTHPC-loaded micelles were studied. The photocytotoxicity of the micellar PS formulations was evaluated on both A431 and HeLa tumor cell lines by a 3-(4,5-dimethylthiazol-2-yl)-5-(3-carboxymethoxyphenyl)-2-(4-sulfophenyl)-2*H*-tetrazolium (MTS) assay. Importantly, blood circulation kinetics and biodistribution of the micelles and the incorporated mTHPC were studied and compared with free mTHPC in A431 tumor-bearing mice using fluorescence intensity measurements and high performance liquid chromatography analysis, respectively, while the biodistribution of micelles was also visualized by 2D fluorescence reflectance imaging to reveal the correlation behavior between the cargo and its host micelles. 

## 2. Materials and Methods

### 2.1. General

The ε-caprolactone (CL), methoxy-poly(ethylene glycol) (mPEG-OH, 2000 g/mol), methanesulfonic acid (MSA, ≥99.0%), pyridine (99.8%), benzyl bromide (98%), and triethylamine (TEA) were obtained from Sigma-Aldrich (Zwijndrecht, The Netherlands). Phosphate-buffered saline (PBS, pH 7.4, containing 11.9 mM phosphates, 137 mM sodium chloride and 2.7 mM potassium chloride) was obtained from Fischer Bioreagents (Bleiswijk, The Netherlands). Standard regenerated cellulose dialysis tubing (Spectra/Por^®^6) with a molecular weight cutoff (MWCO) of 1 kDa was purchased from Spectrumlabs (Rancho Dominguez, CA, USA). Radioimmunoprecipitation assay (RIPA) lysis buffer (10×, 0.5 M Tris-HCl, pH 7.4, 1.5 M NaCl, 2.5% deoxycholic acid, 10% NP-40, 10 mM EDTA) was purchased from Merck KGaA (Darmstadt, Germany). The 2,2-*Bis*-thiomethyl-trimethylene carbonate (TTC) was kindly provided by Professor Zhiyuan Zhong (Soochow University, Suzhou, China). Human epidermoid carcinoma A431 and human cervical carcinoma HeLa cells were obtained from the American Type Culture Collection (ATCC, Manassas, VA, USA). All other solvents and reagents were obtained from Biosolve (Valkenswaard, The Netherlands). The mPEG-OH was azeotropically dried from toluene prior to use. Dichloromethane (DCM, peptide synthesis grade), toluene, and dimethylformamide (DMF) were dried over 4Å molecular sieves (Sigma-Aldrich, Zwijndrecht, The Netherlands) prior to use. All other reagents were used as received. 

The ^1^H/^13^C-NMR spectra of the synthesized monomer and polymers were recorded using a Bruker NMR spectrometer (600 MHz, Bruker, Billerica, MA, USA). The different samples were dissolved in CDCl_3_ at concentrations of approximately 15 mg/mL. Chemical shifts of residual solvent (CHCl_3_: *δ* 7.26 and 77 for proton and carbon spectrum, respectively) were used as the reference lines. Peak multiplicity is designated as s (singlet), d (doublet), t (triplet), and m (multiplet). 

### 2.2. Synthesis of Monomer and Polymers

#### 2.2.1. Synthesis of Monomer

##### Synthesis of Benzyl 2,2-bis(hydroxymethyl)propionate

Benzyl 2,2-bis(hydroxymethyl)propionate was synthesized as previously described (Scheme 1A) [34] and obtained as white needle crystals (47.8 g, yield: 67%) with a melting point of 73 °C (Appendix A). ^1^H-NMR (600 MHz, CDCl_3_): *δ* 7.38 (m, OCOCH_2_***C_6_H_5_***), 5.21 (s, OCO***CH_2_***C_6_H_5_), 3.95-3.72 (m, HO***CH****_2_*C***CH_2_***OH), 1.09 (s, C***CH_3_***). ^13^C-NMR (150 MHz, CDCl_3_): *δ* 175.2,135.6, 128.6, 128.3, 127.8, 68.3, 66.7, 49.2, 17.1. 

##### Synthesis of Benzyl 5-methyl-2-oxo-1,3-dioxane-5-carboxylate

Benzyl 5-methyl-2-oxo-1,3-dioxane-5-carboxylate (i.e., trimethylene carbonate functionalized by benzyl group (TMC-Bz)) was synthesized as previously described (Scheme 1A) [34]. The product was obtained as a white solid (40.6 g, yield: 97%) and further purified by recrystallization from ethyl acetate prior to being used for polymerization. The product after recrystallization was composed of white needle crystals and its melting point shifted from 73 (before recrystallization) to 75 ℃ (Appendix A). ^1^H-NMR (600 MHz, CDCl_3_): *δ* 7.38 (m, OCOCH_2_***C_6_H_5_***), 5.22 (s, OCO***CH_2_***C_6_H_5_), 4.71(m, COO***CH****_2_*C***CH_2_***OCO,), 4.21 (m, COO***CH****_2_*C***CH_2_***OCO), 1.33 (s, OCH_2_C***CH_3_***). ^13^C-NMR (150 MHz, CDCl_3_): *δ* 170.9, 147.4, 134.7, 128.7, 128.2, 72.9, 67.9, 40.2, 17.6.

#### 2.2.2. Synthesis of Polymers

##### 2.2.2.1. Synthesis of P(CL/TMC-Bz)-PEG

A representative procedure for the synthesis of poly(ɛ-caprolactone)-*co*-poly(benzyl 5-methyl-2-oxo-1,3-dioxane-5-carboxylate)-*b*-poly(ethylene glycol) (P(CL/TMC-Bz)-PEG, Scheme 1B) was carried out as previously described [35,36]. Briefly, CL (342 mg, 3.0 mmol), TMC-Bz (750 mg, 3.0 mmol), and mPEG-OH (660 mg, 0.33 mmol) were dissolved in 7 mL dry DCM, followed by addition of MSA (37 mg, 0.39 mmol) with agitation. The reaction was allowed to proceed at 37 °C under N_2_ atmosphere. At predetermined time points, samples were withdrawn from the reaction mixture and analyzed using ^1^H-NMR spectroscopy to monitor monomer conversion in time, and thus the polymerization kinetics of CL and TMC-Bz. After 10 h, TEA (54 µL, 0.39 mmol; equimolar to MSA) was added to terminate the reaction. The cooled reaction solution was dropped into a 20-fold excess of cold diethyl ether (−20 °C) and the precipitate was collected by filtration and dried under vacuum to give the final product (entry 7 in Table 1). 

PCL-PEG (Entries 1 and 5, Table 1) and P(TMC-Bz)-PEG (entry 3, Table 1) block copolymers were synthesized using MSA as the catalyst under the same conditions by polymerization of only CL or TMC-Bz, respectively. 

The polymerization kinetics were determined by monitoring the decrease of peak integrals of methylene of CL at 2.66 ppm and methylene of benzyl group in TMC-Bz at 4.69 ppm. The peak originating from the three methoxy protons of mPEG-OH at 3.37 ppm was used as the reference peak to normalize the integrals. 

##### 2.2.2.2. Synthesis of Bz-PCL-PEG

To obtain benzylated PCL-PEG (Bz-PCL-PEG), the hydroxyl end groups of PCL-PEG were reacted with benzoyl chloride (Scheme 1C), as reported before [37]. In short, 1.2 g of PCL-PEG was dissolved in 6 mL dry DCM, which also contained a 5-fold molar excess of TEA compared to PCL-PEG. This solution was subsequently added dropwise to a solution of 5 equivalents of benzoyl chloride in 3 mL of dry DCM and stirred overnight under a nitrogen atmosphere. Finally, the solvent was removed under reduced pressure and the obtained residue was dissolved in DCM and purified by precipitation in an excess of diethyl ether (−20 °C). The collected precipitate of Bz-PCL-PEG was dried under vacuum and obtained as a white solid. 

### 2.3. Polymer Characterization

The average degree of polymerization (DP) of CL or TMC-Bz in the obtained copolymers was determined by ^1^H-NMR; that is, from the ratio of the integral of the CH_2_ protons of the CL units (1.39 ppm, 2H, CH_2_CH_2_***CH_2_***CH_2_CH_2_) or the CH_2_ protons of benzyl groups in the poly(TMC-Bz) block (5.15 ppm, 2H, O***CH_2_***C_6_H_5_) to the methyl protons of mPEG-OH (3.37 ppm, 3H, ***CH_3_***O), respectively. The number average molecular weight (M_n_) of the block copolymers was calculated from the resulting DP of CL and TMC-Bz units and the molecular weight of the PEG block.

Gel permeation chromatography (GPC) analysis of the synthesized polymers was conducted to determine the number average molecular weight (M_n_), weight average molecular weight (M_w_), and polydispersity (PDI, equal to M_w_/M_n_) of the obtained block copolymers, using two PLgel Mesopore columns (300 × 7.5 mm, including a guard column, 50 × 7.5 mm) coupled with a differential refractive index (RI) detector. Poly(ethylene glycol)s of narrow molecular weights ranging from 430 to 26,100 g/mol were used as calibration standards. The eluent was DMF containing 10 mM LiCl, the elution rate was 1.0 mL/min, and the temperature was set at 65 °C [24].

Differential scanning calorimetry (DSC) was carried out using a Discovery DSC (TA Instruments, New Castle, DE, USA) calibrated with indium. Samples (~5 mg) were heated with a ramp of 3 °C/min up to 150 °C (modulated), annealed for 5 min, cooled down at 3 °C/min to −80 °C (modulated), again annealed for 5 min, and subsequently heated at 3 °C/min up to 150 °C (modulated). Melting temperatures (T_m_) were obtained from the onset of the peaks of the total heat flow and the melting enthalpies (ΔH_m_) were recorded from the total heat flow. Glass transition temperatures (T_g_) are defined as the point of inflection of the step change observed in the reversing heat flow curve. Data for the second heating cycle were recorded.

### 2.4. Synthesis and Characterization of Cy7-Labeled P(CL/TTC)-PEG

For in vivo studies, Cy7-labeled polymers were obtained by first synthesizing a block copolymer containing pendant thiol groups (i.e., poly(ɛ-caprolactone-*co*-2,2-*bis*-thiomethyl-trimethylene carbonate)-*b*-poly(ethylene glycol) (P(CL_18_-TTC_7.5_)-PEG)), and allowing this polymer to react with maleimide-functionalized near-infrared (NIR) fluorophore Cyanine7 (Cy7) (Lumiprobe Corporation, Hannover, Germany), as described previously (Appendix A) [38]. In short, CL was copolymerized with TTC using mPEG-OH as an initiator and MSA as a catalyst (molar ratio of CL/TTC/mPEG-OH was 18/8/1), following the same procedure as described in Section 2.2.2.1 for the synthesis of P(CL/TMC-Bz)-PEG. Subsequently, the disulfide bonds in the resulting P(CL_18_-TTC_7.5_)-PEG were reduced using tris(2-chloroethyl) phosphate (TCEP) to yield free thiol groups, which in turn were used for reaction with Cy7-maleimide (0.64 equivalent per polymer chain) via the thiol-maleimide reaction (Appendix A). After the reaction (4 h at room temperature), the unreacted free thiols were blocked by reaction with maleimide (4 h at room temperature), and subsequently, the unreacted maleimide and unreacted Cy7 were removed by dialysis against THF/water (50:50 volume ratio) for 3 days. The successful coupling of Cy7 with the polymer and complete removal of free Cy7 was demonstrated by GPC analysis, with which the amount of Cy7 in the copolymer was quantified using UV-Vis detection at 755.5 nm, showing 17% coupling efficiency, as reported previously [38]. On average, one polymer chain carried 0.17 Cy7 label. 

### 2.5. Preparation and Characterization of Empty and mTHPC-Loaded Micelles

Empty micelles were prepared by a nanoprecipitation method, as previously described and with a slight modification [39]. In short, 10 mg of block copolymer was dissolved in dimethylsulfoxide (DMSO, 100 µL). After vortexing for 1 min, the mixture was heated up to 70 °C for 5 min to obtain a homogenous solution. This warm solution was cooled down to room temperature and then added dropwise to PBS at 1:9 volume ratio. A homogenous micellar dispersion was formed after gentle shaking, followed by dialysis using tubing (MWCO = 1 kDa) against PBS at room temperature for 12 h. The micellar dispersion obtained after dialysis was filtered through a 0.2 µm syringe filter. The Z-average hydrodynamic diameter (Z_ave_) and polydispersity index (PDI) of the formed micelles after dialysis were determined by dynamic light scattering (DLS) at a fixed scattering angle of 173° and at 25 °C using a ZetaSizer Nano S (Malvern, Surrey, UK). 

The mTHPC-loaded micelles (different loading percentages) were prepared as follows. A certain volume of mTHPC solution in DMSO (10 mg/mL added volume depending on the aimed wt% loading) was added to the weighted polymer, followed by addition of a certain volume of DMSO to obtain a final polymer concentration of 100 mg/mL. Subsequently, the procedures were the same as mentioned above (i.e., the mixture was heated and then added to PBS, followed by dialysis against PBS). The absorbance of the micellar dispersion diluted in DMSO was recorded at 651.5 nm using a UV-2450 Shimadzu spectrophotometer (Kyoto, Japan) and calibration was done using a series of standard solutions of mTHPC in DMSO to determine the mTHPC loading [24]. The loading efficiency (LE) and loading capacity (LC) of mTHPC were calculated using the following Equations (1) and (2), respectively:(1)$$LE\;\left(\%\right)=\frac{mTHPC\;loaded\;\left(mg\right)}{mTHPC\;in\;the\;feed\;\left(mg\right)}\times 100\%$$
(2)$$LC\;\left(\%\right)=\frac{mTHPC\;loaded\;\left(mg\right)}{polymer\;\;used\;\left(mg\right)+mTHPC\;loaded\;\left(mg\right)}\times 100\%$$

### 2.6. Aggregation State of mTHPC

The mTHPC-loaded micelles with different loadings of mTHPC were prepared in PBS, as described in Section 2.5. The micellar dispersions were diluted 10 times in PBS (the final polymer concentration was 1 mg/mL). The fluorescence intensity of mTHPC was recorded using a Jasco FP8300 spectrofluorometer (Tokyo, Japan) at 655 nm (excitation at 420 nm) and plotted against the concentration of mTHPC loaded in the micelles. 

### 2.7. In Vitro Release of mTHPC-Loaded Micelles in Human Plasma 

The in vitro release of mTHPC-loaded micelles with 5 wt% mTHPC loading (prepared in PBS as described in Section 2.5) was studied in human plasma at 37 °C by monitoring the change of fluorescence intensity of mTHPC, as previously reported [24]. Foscan^®^ (i.e., free mTHPC solution in ethanol/propylene glycol (40:60, *w/w*)) was used as a reference. In short, different formulations were added to human plasma at a volume ratio of 1:9. As controls, samples were diluted with PBS (1:9, *v/v*). After incubation at 37 °C, samples were taken at different time points (5 min, and 0.5, 1, 1.5, 2, 3, 5, 8 h) and pipetted into the wells of a 384-well plate to record the fluorescence intensity, as described in Section 2.6.

In addition, samples of Foscan^®^ and micellar mTHPC formulation were taken after incubation with human plasma (1:9, *v/v*) at 37 °C for 5 h, and then 1.5, 2, 4, and 30 times diluted with either human plasma or PBS. After incubation at 37 °C, samples were taken at 0.5, 1, and 2 h and transferred into a 384-well plate to record the fluorescence intensity.

### 2.8. Dark Cytotoxicity and Photo-Cytotoxicity of Empty and mTHPC-Loaded Micelles 

A431 and HeLa cells were cultured in Dulbecco’s modified Eagle’s medium (DMEM) containing glucose (1 g/L for A431 and 4.5 g/L for HeLa) and supplemented with 10% (*v/v*) fetal bovine serum (FBS). The cells were kept in culture at 37 °C in a humidified 5% CO_2_ atmosphere. 

Dispersions of empty and mTHPC-loaded micelles (with various loadings) were prepared in PBS (10 mg/mL polymer), as described in Section 2.5. The stock dispersions were diluted in DMEM medium (2.5 and 5 times dilution for empty micelles and 10 times dilution for the mTHPC-loaded micelles) prior to cell exposure for evaluation of their cytotoxicity (empty micelles), dark cytotoxicity, and photo-cytotoxicity (mTHPC-loaded micelles) on A431 and HeLa cells. 

The cells were seeded into 96-well plates at a density of 6000 A431 cells/well or 5000 Hela cells/well and incubated overnight at 37 °C and 5% CO_2_. Subsequently, the medium (100 μL) in the wells was replaced by 100 μL of the above-described empty or mTHPC-loaded micellar dispersions. To evaluate the photocytotoxicity of the different micellar mTHPC formulations, the cells were incubated first in the dark for 7 h at 37 °C and 5% CO_2_ with the different formulations. Next, the medium with the formulations was removed and cells were washed three times with DMEM medium. Subsequently, the cells in 100 μL of fresh DMEM were illuminated for 10 min with a light intensity of 3.5 mW/cm^2^ (corresponding to 2.1 J/cm^2^), using a homemade device consisting of 96 LED lamps (650 ± 20 nm, 1 LED per well), and then incubated overnight at 37 °C and 5% CO_2_. Finally, cell viability was measured as described before [24], by recording the absorbances of the different wells at 490 nm after the cells were exposed for approximately 1 h to a CellTiter 96^®^ AQ_ueous_ One Solution (Promega, Leiden, The Netherlands) containing 3-(4,5-dimethylthiazol-2-yl)-5-(3-carboxymethoxyphenyl)-2-(4-sulfophenyl)-2*H*-tetrazolium (i.e., MTS assay). 

For determination of the cytotoxicity of empty polymeric micelles (without mTHPC loading) and dark toxicity of the above described mTHPC-loaded micelles, the cells were incubated with the different formulations in the dark for 24 h at 37 °C and 5% CO_2,_ and subsequently the cell viability was determined directly (without irradiation) by the MTS assay after washing off the formulations [24].

### 2.9. In Vivo Studies of mTHPC and Micelles in A431 Tumor-Bearing Mice

For the in vivo studies, P(CL_9.1_-TMC-Bz_7.7_)-PEG and Bz-PCL_17.6_-PEG micelles loaded with mTHPC (0.6 wt% loading) were used and prepared as described in Section 2.5, except that micelles were labeled by mixing the block copolymer with Cy7-labeled P(CL_18_-TTC_7.5_)-PEG (at a ratio of 98.5% to 1.5% *w/w*). A formulation of free mTHPC was prepared by 1:1 dilution of a 120 µg/mL mTHPC stock solution in Foscan^®^ solvent (i.e., ethanol/propylene glycol, 40/60 *w/w*) with PBS (final mTHPC concentration was 60 µg/mL, corresponding to an equal dose of injected mTHPC as micellar formulation. 

The animal experiments were approved by the Central Animal Experiments Committee (approval number #AVD108002016544) and the Animal Welfare Body Utrecht (approval number WP# 544-2-04). Female Balb/c nude mice, weighing 20–28 g, were purchased from Envigo (Horst, The Netherlands). Mice were housed in ventilated cages at 25 °C and 55% humidity under natural light/dark conditions. Food and water were provided ad libitum during the entire study. Mice were inoculated with 1 × 10^6^ A431 cells suspended in 100 μL PBS subcutaneously into the right flank. When the tumors reached an approximate size of 100–300 mm^3^ (between 7 and 14 days after injection of the tumor cells), mice were included in the studies. Tumors were measured using a digital caliper. The tumor volume V (in mm^3^) was calculated using the equation V = (π/6)LS^2^, in which L is the largest and S is the smallest superficial diameter [30].

#### 2.9.1. Circulation Kinetics

Three groups of tumor-bearing mice (*n* = 4-6 per group) were intravenously (i.v.) injected via the tail vein with free mTHPC formulation (i.e., mTHPC dissolved in diluted Foscan solvent (ethanol/propylene glycol/PBS 20:30:50 *v/v/v*)) or Cy7-labeled mTHPC micelles, respectively, at injection doses of 300 µg mTHPC/kg, corresponding to ~6 µg mTHPC in ~120 μL per individual mouse with average body weight of ~25 g.

Blood samples were collected in tubes with EDTA anticoagulant via submandibular puncture (~60 μL) from mice at 1 min (100% injection control), then at 1 and 2 h, and via cardiac puncture (~200 μL) at 4 or 24 h post-injection. For the latter, mice were killed through cervical dislocation while under deep isoflurane anesthesia. The collected blood samples were centrifuged at 1000× *g* for 15 min at 4 ℃. The plasma supernatant was collected, extracted using acetonitrile/DMSO (4:1, *v/v*), and analyzed by high-performance liquid chromatography (HPLC) and a LI-COR Odyssey imaging system to quantify the amount of mTHPC and Cy7-labeled micelles, respectively, as previously described [38]. Plasma concentration curves were analyzed by non-compartmental analysis with the PKSolver add-in for Microsoft Excel [40].

#### 2.9.2. Biodistribution

The mice were sacrificed 4 (3–6 animals per group) or 24 h (3–6 animals per group) after i.v. administration of the formulations. Tumors and a panel of organs (spleen, liver, lung, heart, kidney, skin, femur, and brain) were excised and imaged ex vivo by 2D fluorescence reflectance imaging (FRI) using a Pearl Trilogy imager from LI-COR and then stored at −80 °C until further processing for quantification. Organs and tumors from three untreated animals were used as controls. 

To quantify the content of mTHPC and micelles in the tumors and the different organs, the excised tissue samples were treated as follows. First, 100 μL of RIPA lysis buffer was added to 100 mg of sliced tissues or organs. The mixture was homogenized by a tissue grinder (Percellys 24) at a speed of 6000/s for 60 s (for femur samples, 6000/s for 180 s) and the homogenate was subsequently aliquoted. To determine the mTHPC concentration in the samples, 1 volume of an aliquot of the homogenate (30 μL) was mixed with 120 μL of acetonitrile/DMSO (4:1 *v/v*) and vortexed for 1 min. The mixture was then centrifuged at 15,000× *g* for 10 min. Next, 50 μL of the obtained supernatant was injected into the HPLC system consisting of a Waters X Select Charged Surface Hybrid (CSH) C18 3.5 μm 4.6 × 150 mm column coupled with a fluorescence detector set at λ_ex_ 420 nm and λ_em_ 650 nm to analyze mTHPC concentration [38]. The mobile phase was 0.1 % trifluoroacetic acid in acetonitrile/water (60:40, *v/v*) at a flow rate of 1 mL/min. The measuring range was from 0.005 to 4 μg/mL and the detection limit was 5 ng/mL. Calibration curves were obtained from a series of standard solutions of mTHPC in DMSO, to which 45 μL of the corresponding homogenized tissue samples obtained from control mice (i.e., not treated with any formulations) was added, followed by mTHPC extraction using acetonitrile/DMSO (4:1, *v/v*) and HPLC analysis.

To determine the concentration of Cy7-labeled micelles in the tissue homogenates, 1 volume of another aliquot of the homogenate (30 μL) was vortex-mixed with 2 volumes of RIPA lysis buffer (60 μL) for 1 min. The fluorescence of Cy7 in the mixture (20 μL) was detected at the 800 nm channel (i.e., λ_ex_ 785 nm and λ_em_ 820 nm), using a LI-COR Odyssey scanner imaging system, and a calibration curve was obtained using samples with different concentrations of Cy7-labeled copolymer in a mixture of RIPA buffer (1 volume) and the corresponding homogenized tumor or organ samples (2 volumes) obtained from non-treated mice. 

It is noted that skin accumulation of different formulations was not included, as grinding of the skin was more problematic in our preliminary test, which could lead to unreliable quantification. In addition, a study by Bovis et al. suggested that mTHPC accumulation in the skin was limited [8].

### 2.10. Statistical Analysis

Statistical analysis was done by GraphPad Prism 8.3.0 software. Two-way analysis of variance (ANOVA) was used to determine the statistical significance of biodistribution among different mTHPC formulations. A value of *p* < 0.05 was considered significant. Statistical significance is depicted as * *p* ˂ 0.05, ** *p* ˂ 0.01, *** *p* ˂ 0.001.

## 3. Results and Discussion

### 3.1. Synthesis and Characterization of Monomer and Polymers

To synthesize benzyl 5-methyl-2-oxo-1,3-dioxane-5-carboxylate (i.e., trimethylene carbonate functionalized by benzyl group, TMC-Bz), 2,2-bis(hydroxymethyl)propionate was first benzylated by reaction with benzyl bromide, followed by reaction with triphosgene (Scheme 1A). After purification, TMC-Bz was obtained in a high yield (97 %) as a white crystalline solid. The structures of the intermediate product and TMC-Bz were confirmed by ^1^H/^13^C NMR spectroscopy (Appendix A). 

The block copolymers synthesized by the ring-opening polymerization of CL or TMC-Bz, initiated by mPEG-OH and catalyzed by MSA using different CL/TMC-Bz/initiator molar ratios (Scheme 1B), were obtained as white solids in a yield of ~65%, and their chemical structures were characterized by ^1^H NMR (Appendix A). The characteristics of the obtained block copolymers are shown in Table 1. The compositions of the resulting block copolymers as determined by ^1^H-NMR fit well with those expected from the ratios of the monomers in the feed. GPC analysis shows narrow molar mass distributions (M_w_/M_n_ ≤ 1.1), suggesting the absence of significant side reactions, such as transesterification [35,41,42,43]. For the ring-opening copolymerization of CL and TMC-Bz, similar polymerization rates of the two monomers were observed, as shown in Figure 1A for a CL/TMC-Bz/mPEG-OH feed molar ratio of 9:9:1 (i.e., entry 7, Table 1). The ^1^H-NMR spectrum of the resulting block copolymer (Figure 1B) displayed three groups of peaks in the ester region at 3.90–4.40 ppm, corresponding to the three kinds of CH_2_O-carbonyl linkages in the different diad structures that are present in the poly(ester-carbonate) block (i.e., TMC-Bz-TMC-Bz, CL-TMC-Bz, and CL-CL, respectively) [42,44]. This demonstrates a random distribution of CL and TMC-Bz in the resulting P(CL/TMC-Bz)-PEG copolymers (entires 2, 4, 6, and **7**, Table 1) which is in good agreement with the equal reactivity of the two monomers. 

It is noted that the same characteristics of PCL_17.6_-PEG and its terminal benzyl derivative (i.e., Bz-PCL_17.6_-PEG) (entries 4 and 8, Table 1) indicate that loss of materials and undesired side reactions did not occur during the process of post-modification of the end groups (Scheme 1C). The assignments of the corresponding NMR peaks of Bz-PCL_17.6_-PEG (Appendix A) were in line with those reported previously [37]. In addition, the ^1^H-NMR spectrum of Bz-PCL_17.6_-PEG (Appendix A) shows that the integral ratio of the protons of the terminal Bz group at 7–8 ppm to the terminal OCH_3_ group of PEG at 3.37 ppm was 5:3 (corresponding with a molar ratio of Bz/OCH_3_ of 1:1), demonstrating that all polymer chains carry a terminal benzyl end group.

The thermal properties of the obtained block copolymers were investigated by DSC (Table 1, representative thermograms are shown in Appendix A). All block copolymers showed only one T_m_ at around 40 ^o^C, close to that of mPEG-OH (48 °C), which is in accordance with previous data [24,45]. For the copolymers, crystallinity (i.e., △H_m_) of PEG corrected for its weight fraction in the corresponding polymer chains was well in accordance with that of the mPEG-OH (measured △H_m_ was 182 J/g) (Appendix A), demonstrating that in the solid state, PEG and the polyester/carbonate blocks were phase-separated in the crystalline PEG domain and amorphous P(CL-TMC-Bz) (entries 2, 4, 6 and 7, Table 1) with T_g_’s, ranging from −47 to −11 °C. These T_g_’s can be described by the Fox equation (Appendix A), suggesting a random distribution of CL and TMC-Bz in the polymer chains, which is in line with that observed from NMR analysis (Figure 1B). These amorphous domains are obviously not miscible with PEG. The PCL-PEG block copolymers (entries 1, 5, and 8, Table 1) were almost fully crystalline: both PCL and PEG have their T_m_’s around 45 °C [45], while the P(TMC-Bz)-PEG (entry 3, Table 1) block copolymer had T_g_ at −12 °C, which resulted from the P(TMC-Bz) block and is in reasonable agreement with the T_g_ of PTMC_80_ (−25 °C) [35]. 

### 3.2. Preparation and Characterization of Polymeric Micelles

Empty polymeric micelles were prepared at a polymer concentration of 10 mg/mL using a nanoprecipitation method by dropping a polymer solution in DMSO into an excess of PBS followed by dialysis. DLS measurements (Figure 2A) demonstrate that PCL_9_-PEG (entry 1, Table 1) without aromatic rings formed directly after dialysis micelles with a size of 27 nm and a PDI of ~0.3, which was in the range observed previously for low molecular weight oligolactates-*b*-PEG and oligocaprolactones-*b*-PEG (M_n_ < 1.8 kDa) with unmodified hydroxide end groups [37,45]. Importantly, the size and PDI of the PCL_9_-PEG micelles (Appendix A) further increased to ~80 nm (PDI ~0.4) during 24 h storage at room temperature, in combination with an increased derived count rate from 2000 to 6000 (Appendix A), demonstrating formation of aggregates, pointing to a low colloidal stability. Similarly, the larger PCL_17.6_-PEG micelles (entry 5, Table 1) showed a size of ~600 nm with PDI of ~0.7 even directly after dialysis. On the other hand, micelles from the corresponding benzyl-terminated polymer (entry 8, Table 1) showed substantially decreased size and size distribution (23 nm with PDI of 0.3, Figure 2A). This result is in accordance with our previous studies, in which it was demonstrated that micelles based on PCL_9_-PEG with a benzyl end group showed smaller size (17 nm), lower PDI (<0.1), and better stability than the non-benzylated polymer micelles [24,38]. 

Figure 2A shows that the different micelles based on P(CL/TMC-Bz)-PEG or P(TMC-Bz)-PEG block copolymers with different amounts of pendant benzyl groups (Entries 2–4, 6 and 7, Table 1) had small hydrodynamic diameters that slightly increased with the increasing chain length of the hydrophobic blocks (from 17-23 nm) with PDIs < 0.1 (Figure 2A). It is noted that the different aromatic substituted micelles, including Bz-PCL_17.6_-PEG micelles, retained their small sizes and low PDIs for at least 7 days in PBS at room temperature (Figure 2A, green columns). Overall, these results demonstrate that aromatic groups, regardless of their positions and the amounts in the polymer chains, stabilized the cores of micelles based on PCL-PEG. In addition, from the perspective of in vivo application, the relatively small sizes of these stable micelles are expected to favor both their accumulation in tumors through the EPR effect [21,22] and subsequent penetration into the tumor with uniform distribution, being crucial factors for anti-tumoral efficacy of nanomedicines [46,47,48]. 

The P(CL/TMC-Bz)-PEG and (Bz-)PCL-PEG-based micelles were loaded with mTHPC with loading efficiencies of ≥80%, as determined by UV-Vis analysis (Figure 2B, solid lines), independent of the composition of copolymers used, which led to a linear increase of loading capacity up to ~8 wt % with increasing mTHPC feed (Figure 2B, dash lines). 

### 3.3. Aggregation State of mTHPC in Polymeric Micelles

The influence of the number of aromatic rings in the copolymers on the aggregation state of mTHPC in the different micelles was determined by measuring the quenching of its fluorescence [24,33,49]. For example, Yang et al. showed that the quenching concentration of a phthalocyanine (i.e., Si(sol)_2_Pc) in aromatic-substituted thermosensitive micelles (0.45 mg/mL polymer) increased around 300 times to 16 µg/mL compared to that in corresponding non-aromatic micelles (~0.05 µg/mL) [33]. As discussed in Section 3.2, polymeric micelles prepared from non-modified PCL-PEG copolymers showed low stability. Therefore, only the aromatic micelles with either benzyl-modified end groups or differing contents of benzyl pendant groups in the polymer chains were loaded with increasing amounts of mTHPC and the fluorescence intensity of mTHPC was measured. Figure 3A shows that the different mTHPC-loaded micelles exhibited similar fluorescence profiles. The fluorescence of mTHPC-loaded micelles increased almost linearly with increasing mTHPC loading from 0.06 to 0.5% *w/w* (i.e., ~0.6 to ~5 µg/mL mTHPC; the final polymer concentration was 1 mg/mL). However, at higher loading percentages, mTHPC showed a rapid decrease in fluorescence intensity with increasing mTHPC loading, suggesting that fluorescence quenching resulting from mTHPC in the micelles occurred at ~10 µg/mL (corresponding to 1 wt% mTHPC loading), in agreement with our previous work on mTHPC-loaded Bz-PCL-PEG micelles [24]. A higher maximum fluorescence was observed when mTHPC was loaded in P(CL/TMC-Bz)-PEG micelles with a high content of aromatic rings (i.e., molar ratio of PTMC-Bz/PCL in polymer chains >0.3; Figure 3A, cyan, pink, yellow, and purple lines), because of the observed slightly later onset of quenching.

### 3.4. In Vitro Release of mTHPC from Micelles in Human Plasma

The quenched state of the fluorescence when mTHPC is present in the micelles above 0.5% *w/w* loading (see Figure 3A) was used to investigate the in vitro stability of the mTHPC micellar formulations in human plasma [24]. When mTHPC is released from the micelles, this should lead to less quenching (i.e., increase in fluorescence intensity) as the local concentration inside the micelles is decreased, which is reinforced due to fluorescence of the free mTHPC likely bound to plasma proteins. For this purpose, the fluorescence of different micellar dispersion with 5wt% mTHPC feed loading, as well as mTHPC in its free form (Foscan^®^), was assessed in human plasma over time at 37 ℃ (Figure 3B). As expected, the fluorescence of mTHPC-loaded micelles upon 10× dilution in PBS did not change in time over 8 h at 37 °C (Appendix A). Upon 10× dilution in plasma, Foscan^®^ gave the highest immediate fluorescence, which increased slightly further during the first 30 min incubation and then remained stable at a value of ~2600 arbitrary units (a.u.); Figure 3B, black line). For all micellar mTHPC formulations (Figure 3B), substantial increase of fluorescence was observed to different levels within the first 1 h of incubation, and then the fluorescence intensities leveled off, reflecting different degrees of release of mTHPC from the micelles [24]. Interestingly, the plateau fluorescence intensities of mTHPC loaded in non-aromatic PCL_9_-PEG or PCL_17.6_-PEG micelles and Bz-PCL_17.6_-PEG micelles (Figure 3B, blue, red, and green lines) were identical to that of Foscan^®^ (at ~2600 a.u.), suggesting that the release was almost complete. This suggests that incorporating a single terminal aromatic group did not improve the retention of mTHPC in micelles in plasma, despite the enhanced colloidal stability of those micelles in PBS (see Figure 2A). In contrast, when mTHPC was loaded in micelles based on P(CL/TMC-Bz)-PEG or P(TMC-Bz)-PEG (i.e., with pendant aromatic rings), fluorescence intensities increased to lower levels than those loaded in the micelles from non-aromatic and chain-end modified polymers with a similar total length of hydrophobic blocks (Figure 3B, pink and cyan lines vs. blue line, or purple and dark blue lines vs. red and green lines). When the total length of hydrophobic blocks was kept more or less constant, the fluorescence intensity of mTHPC leveled off from 2600 a.u. (red line) to 1500 a.u.(dark blue line) and 1300 a.u. (purple line) as the content of TMC-Bz units increased from 0 to 4 and 8 units per polymer chain, respectively. On the other hand, when the degree of polymerization of TMC-Bz was ~8, the plateau fluorescence intensities of mTHPC were 2200 a.u. (pink line), 1800 a.u. (yellow line), and 1300 a.u. (purple line), with increasing total degree of polymerization (and thus molecular weight) of hydrophobic blocks from 8 to 14 and 17 units, respectively. Taken together, in plasma, mTHPC was most efficiently retained in micelles consisting of P(CL_9.1_-TMC-Bz_7.7_)-PEG, which have the most TMC-Bz units and the longest hydrophobic blocks (Figure 3B, purple line). These results convincingly show that the retention of mTHPC in micelles was improved by pendant aromatic rings on the polymer backbone, while the extent of improvement depended on both the hydrophobic chain length and the number of aromatic moieties. The reported findings can be explained by the increased hydrophobic interactions and π-π stacking between the hydrophobic blocks of P(CL_9.1_-TMC-Bz_7.7_)-PEG and mTHPC, which can to some extent prevent the extraction of the cargo from the micelles or the micellar destabilization resulting from the binding of mTHPC or amphiphilic polymer molecules with proteins such as albumin and lipoproteins present in human plasma.

To establish whether the observation of different plateau fluorescence levels of mTHPC-loaded micelles containing TMC-Bz is the result of simple equilibrium partitioning of mTHPC between micelles and plasma proteins, mTHPC-loaded P(CL_9.1_-TMC-Bz_7.7_)-PEG micelles, after incubation with human plasma for 5 h, were further diluted with human plasma or PBS in different proportions and compared with Foscan^®^ samples treated the same way. The solid black lines in Figure 4 show that upon further dilutions in human plasma, regardless of the dilution factors, the fluorescence of Foscan^®^ samples was constant for 2 h and comparable to that observed upon dilution in PBS (broken black lines). The latter observation suggests that the amount of proteins present before further dilution was already sufficient to solubilize the amount of mTHPC that was present. When micelles were first incubated with plasma and subsequently diluted with PBS, the fluorescence intensity of mTHPC that was released from the micelles did not change in time (Figure 4, broken purple lines) and remained lower than diluted Foscan^®^, except for the highest dilution factor, which may point to further extraction of the PS from the micelles. Upon 1.5× dilution of the micelles with plasma instead of with PBS, the fluorescence intensity of mTHPC remained constant over time, which was again lower than the fluorescence intensity of the Foscan^®^ sample (compare the solid purple line with the solid black line in Figure 4A). However, with further increase of the dilution factor of micelles in plasma to 2 and 4 times, the fluorescence of mTHPC in micelles showed an increase during the first 1 h incubation and then leveled off at different values (Figure 4B,C, solid purple lines). Upon 2x dilution in plasma, the plateau fluorescence was still lower than that observed from the corresponding Foscan^®^ sample, while in the case of 4x dilution, the plateau fluorescence intensity of micellar mTHPC reached an identical level as observed in the Foscan^®^ sample in plasma. Also, upon extensive dilution in plasma (30×), Foscan^®^ and micellar mTHPC exhibited similar fluorescence levels (Figure 4D, purple lines). These results indicate that complete release of mTHPC from P(CL_9.1_-TMC-Bz_7.7_)-PEG micelles was achieved when more plasma (≥4 times) was added. In other words, the equilibrium partitioning of mTHPC between micelles and plasma depends not only on the strength of the interaction with the polymer, but also on the ratio between micelles and plasma. 

### 3.5. Dark Cytotoxicity and Photo-Cytotoxicity of Empty and mTHPC-Loaded Polymeric Micelles 

Cell viability assays were performed on A431 and HeLa tumor cell lines to assess the cytocompatibility of the empty micellar formulations and to determine the dark toxicity and photo-toxicity of mTHPC-loaded micelles. These experiments were carried out with micelles based on Bz-PCL_17.6_-PEG and three micelles based on polymers with pendant aromatic groups (i.e., P(CL/TMC-Bz)-PEG) with different PCL/PTMC-Bz ratios. Although all used micelles showed required colloidal stability in PBS (Figure 2A), the latter displayed better stability in human plasma (Figure 3B), as shown in Section 3.2 and Section 3.4. Appendix A shows that both A431 and HeLa cells incubated with empty micelles retained their viability at polymer concentrations up to 4 mg/mL, demonstrating that the different micelles have excellent cytocompatibility. As shown in Figure 5A,B, mTHPC-loaded in these different micelles, even with the highest mTHPC concentration up to 81 µg/mL, showed no cytotoxicity on A431 and HeLa cells after incubation with cells in the dark for 24 h. However, we showed before that free mTHPC (i.e., Foscan^®^) was toxic to cells without illumination at mTHPC concentrations higher than 50 µg/mL after 7 h and 20 µg/mL after 24 h [38]. This markedly decreased dark cytotoxicity of mTHPC by loading in the micelles is in line with previous observations using other micellar and liposomal formulations of mTHPC [50,51,52]. 

The observed lack of dark toxicity could have been the result of no uptake of the micelles. However, our previous studies showed that PCL-PEG-based micellar mTHPC formulations and free mTHPC can both be effectively taken up by cells [38]. Indeed, photocytotoxicity studies (Figure 5C,D) show that the different micellar mTHPC formulations were able to induce cell killing upon irradiation and exhibited comparable cytotoxic effects on A431 and HeLa cells (half-maximal effective concentration (EC_50_) values were approximate 4–11 µg/mL for A431 cells and 5–13 µg/mL for HeLa cells, respectively; see Table 2). The EC_50_ value of Bz-PCL_17.6_-PEG on A431 cells was lower than previously observed when mTHPC was loaded in micelles from slightly bigger benzyl-terminated Bz-PCL_23_-PEG polymers (~35 µg/mL) under the same conditions [38]. A similar trend (i.e., increasing photocytotoxicity with decreasing polymer molecular weight) was observed before and attributed to faster intracellular degradation of smaller PCL-PEG block copolymers, and thus faster release of mTHPC after internalization by the cells [24]. It is worth noting that the observed EC_50_ values of these micellar mTHPC formulations on A431 and HeLa cells were slightly higher than free mTHPC (~1.5 μg/mL, Table 2), probably related to the less efficient cellular internalization of PEGylated micelles [53,54,55] or the relatively time-consuming degradation of polymers for release and activation of the PS [24].

### 3.6. Pharmacokinetics and Biodistribution of mTHPC in its Free Form and Loaded in Cy7-Labeled Bz-PCL_17.6_-PEG and P(CL_9.1_-TMC-Bz_7.7_)-PEG Micelles in A431 Tumor-Bearing Mice

The circulation time and biodistribution of mTHPC-loaded micelles were studied using mice bearing human A431 tumor xenografts of 100–300 mm^3^. NIR fluorescence of Cy7-labeled micelles and mTHPC concentrations as measured in plasma and organ and tumor homogenates, thus addressing both polymer and photosensitizer contents of the samples. Micelles consisting of Bz-PCL_17.6_-PEG and P(CL_9.1_-TMC-Bz_7.7_)-PEG were chosen because both copolymers have comparable chain lengths of hydrophobic blocks (~18 units of either CL with the final unit capped with a benzyl group, or CL plus TMC-Bz) but different contents of aromatic units. Importantly, both micellar PS formulations showed similar phototoxicity on cells (Figure 5C,D) but markedly different release behaviors in vitro (Figure 3B). 

The plasma concentrations were determined after intravenous administration of free mTHPC or Cy7-labeled micelles loaded with mTHPC into mice via the tail vein. As reported in our previous study [38], administration of free mTHPC (i.e., mTHPC dissolved in diluted Foscan solvent (ethanol/propylene glycol/PBS 20:30:50 *v/v/v*) inflicted side effects such as tachypnea, passiveness immediately post-injection, and loss of body weight (~1 g on average) within 24 h. However, none of the micellar mTHPC treated mice showed any (short term) side effects during or after their administration, suggesting micellar formulations at the injected polymer dose (~1 mg) are well tolerated for in vivo applications.

The analysis of Cy7 levels in plasma (Figure 6A) showed that the elimination rate of P(CL_9.1_-TMC-Bz_7.7_)-PEG micelles was significantly slower as compared to that of Bz-PCL_17.6_-PEG micelles (e.g., 55% vs 5% of the injected dose (ID) remaining in plasma after 1 h). Non-compartment analysis was used to determine the pharmacokinetic parameters of both micelles, including terminal half-life, area under the curve (AUC), and clearance. As shown in Table 3**,** Bz-PCL_17.6_-PEG and P(CL_9.1_-TMC-Bz_7.7_)-PEG micelles significantly differed in AUC values, and consequentially had large differences in distribution and clearance volumes. The differences in pharmacokinetic parameters primarily relate to the initial phase of the plasma curves (i.e., before 1 h); most likely they relate to differences in stability of the micelles, indicating more rapid dissociation of Bz-PCL_17.6_-PEG micelles into unimers as compared to P(CL_9.1_-TMC-Bz_7.7_)-PEG micelles. This is most likely attributed to relatively stronger π-π stacking between the polymer chains in the latter micelles [30,56]. 

Figure 6B shows the plasma concentrations of mTHPC after administration of either free mTHPC or the micellar formulations. It is clear that free mTHPC and mTHPC loaded in micelles displayed comparable mTHPC levels and showed similar decay profiles in the circulation (Figure 6B). A rapid initial elimination was observed, for which half-lives (t½-α) of ~0.5 h or less were estimated. Non-compartmental analysis of the mTHPC curves provided terminal half-lives (t½-β), AUCs, and derived pharmacokinetic parameters, such as distribution volumes and clearance. It can be concluded from the results presented in Table 3 (bottom part) that those pharmacokinetic parameters were comparable between both free mTHPC and micellar mTHPC formulations. In addition, significant differences of pharmacokinetic parameters were observed between both Bz-PCL_17.6_-PEG micelles and the loaded mTHPC, indicating that PS and polymer dissociated rapidly upon their injection into the circulation. On the other hand, for P(CL_9.1_-TMC-Bz_7.7_)-PEG micellar formulation, pharmacokinetic parameters of mTHPC (Table 3, bottom part) were tightly associated with those derived from Cy7 analysis (Table 3, top part). However, due to coincidently similar data observed for free mTHPC and Cy7, it is difficult to conclude based on these parameters whether mTHPC was released from P(CL_9.1_-TMC-Bz_7.7_)-PEG micelles or retained in the micelles. However, the in vitro release study in human plasma (Figure 3B, purple line and Figure 4) suggests that premature release of mTHPC from micelles most likely also occurred in the circulation for these micelles. Similarly, it was reported that paclitaxel loaded in thermosensitive micelles containing aromatic HPMAm-Bz units in the hydrophobic blocks exhibited a similar pharmacokinetic profile as when loaded in micelles without aromatic groups and as compared to free paclitaxel, despite the significantly improved in vitro stability and drug retention by π-π stacking [30,56]. Additionally, paclitaxel loaded in thermosensitive HPMAm-lactate-based micelles also showed the same pharmacokinetic data as the drug in its free form [57]. In addition, premature cargo release was also observed previously in various liposomal mTHPC formulations and other mTHPC-loaded micelles [24,58]. This release can be attributed to the high binding affinity of mTHPC with plasma (lipo)proteins, leading to mTHPC redistribution from intact micelles to these plasma components [59,60,61]. It is noted that the relatively long circulation time of (released) mTHPC in the β phase is probably due to released mTHPC that subsequently binds to lipoproteins, which can act as endogenous carriers for mTHPC [62,63].

Tumors and organs were excised from mice that were sacrificed at 4 and 24 h after i.v. injection and Cy7-labeled micelles deposited in these tissues were visualized using 2D fluorescence reflectance imaging (FRI) and subsequently quantified by fluorescence intensity. It is noted that before taking out the organs, cervical dislocation followed by a cardiac puncture was performed for all mice, through which a substantial volume of blood was drawn from the animal in order to minimize the background signal from residual blood in the tissues. As shown in Figure 7, ex vivo images indicate that both Bz-PCL_17.6_-PEG and P(CL_9.1_-TMC-Bz_7.7_)-PEG micelles were found to primarily accumulate in liver, followed by kidneys and spleen. Different organ distribution profiles were observed for the two micellar formulations. As an explanation, Bz-PCL_17.6_-PEG micelles accumulated in liver and kidneys at 4 h and while the accumulation in liver decayed, kidney accumulation persisted upon 24 h after administration. Accumulation of P(CL_9.1_-TMC-Bz_7.7_)-PEG micelles in the kidneys was less pronounced and these micelles showed prolonged residence up to 24 h in the liver, and to a lesser extent in the spleen. In line with the imaging results, the quantified fluorescence intensities (Figure 8) show that the prominent liver accumulation of both micelles (Figure 8A) was observed to be ∼15% ID/g at 4 h, in line with previous reports [30,64], most likely due to their clearance by the mononuclear phagocytic system (MPS) [65,66,67]. However, 24 h post-injection, Bz-PCL_17.6_-PEG micelles decreased significantly to approximately 5% ID/g liver, while P(CL_9.1_-TMC-Bz_7.7_)-PEG micelles remained at similar levels in the liver as observed at 4 h. Additionally, the residence of the P(CL_9.1_-TMC-Bz_7.7_)-PEG micelles in the spleen was significantly longer than Bz-PCL_17.6_-PEG micelles (4.6% vs 1.4% ID/g after 24 h, Figure 8D). These results suggest better stability of the P(CL_9.1_-TMC-Bz_7.7_)-PEG micelles than the micelles from Bz-terminated polymers, most likely due to strong π-π stacking resulting from the multiple Bz groups per polymer chain in the micelles of the former. Cy7 levels from Bz-PCL_17.6_-PEG and P(CL_9.1_-TMC-Bz_7.7_)-PEG micelles could be detected in kidneys with ~6% and ~4% ID/g (Figure 8B) at 4 h, respectively, which decreased slightly at 24 h. The observed kidney accumulation was most likely due to dissociation of Cy7-labeled unimers from the micelles, which have a molecular size below the threshold of glomerular filtration [65]. 

Interestingly, although no micelles were detected in the circulation after 24 h (Figure 6), accumulation of P(CL_9.1_-TMC-Bz_7.7_)-PEG micelles in tumors increased from 0.6 % ID/g at 4 h to approximate 1% ID/g at 24 h (Figure 8C). In contrast, no tumor accumulation of Bz-PCL_17.6_-PEG micelles was found at 24 h. This suggests that after 4 h, the remaining P(CL_9.1_-TMC-Bz_7.7_)-PEG micelles in the circulation (15 % ID, Figure 6) were (at least partly) intact and progressively accumulated in tumors via overtime EPR effect. In addition, P(CL_9.1_-TMC-Bz_7.7_)-PEG micelles accumulated in significantly higher amounts in the femur at 4 and 24 h than Bz-PCL_17.6_-PEG micelles (Figure 8G). This also indicates the higher amount of intact micelles present in the circulation, since it was previously reported that relatively small nanoparticles (generally below 60 nm) tend to accumulate in MPS-enriched bone marrow [68,69].

To study a possible correlation of biodistribution between mTHPC and micelles, Figure 9 displays the biodistribution in tumors and a panel of organs of mTHPC that was injected in free form and using Bz-PCL_17.6_-PEG or P(CL_9.1_-TMC-Bz_7.7_)-PEG formulations. The first conclusion is that the biodistribution of micellar mTHPC was similar to that of free mTHPC. The highest mTHPC levels were seen in liver and spleen, followed by lungs (Figure 9A,D,E). These are all tissues containing reticuloendothelial cells with a rich MPS, which are known to preferentially accumulate photosensitizers [8,70,71,72]. The accumulation of mTHPC in the liver at 4 h was similar to the accumulation of polymers in the liver (i.e., ∼15% ID per gram), which then decreased significantly to about 5% ID/g at 24 h (Figure 9A). Higher accumulation of mTHPC in lungs was observed after 4 h than after 24 h (Figure 9E), most likely due to the blood fraction (containing mTHPC) after 4 h present in the excised lungs of non-perfused mice. However, in spleen and lungs, higher accumulation of mTHPC (~8-15% ID/g, Figure 9D,E, green and purple columns) was detected than that of the corresponding host micelles (≤5% ID/g, Figure 8D,E). The values of mTHPC accumulation were consistent with those reported for mTHPC when dosed using different liposomal formulations [58,72,73]. It is noted that although similar mTHPC content was detected per gram for the three organs, the total mTHPC accumulation in liver was significantly higher than that in spleen and lungs (e.g., ~15% ID per liver vs. ~1% ID per spleen or lung at 4 h; Appendix A) due to the high weights of liver samples (~1 g on average). Apart from liver, spleen, and lungs, regardless of the timepoints, similar amounts of mTHPC (~2–3% ID/g) were accumulated in tumor, kidney, heart, and femur (Figure 9B,C,F,G), suggesting unspecific biodistribution of mTHPC. Accumulated mTHPC levels resulting from being loaded in Bz-PCL_17.6_-PEG and P(CL_9.1_-TMC-Bz_7.7_)-PEG micelles in these organs and tissues (Figure 9C,F,G, green and purple columns) were higher as compared to the corresponding host micelles (<1%) (Figure 8C,F,G), except those in the kidney (Figure 8B and Figure 9B). This loose association of the distribution pattern between mTHPC and the host micelles confirmed the fast release of mTHPC from both micelles. It is noted that neither the PS nor the micelles showed disposition in the brain (Figure 8H and Figure 9H), which is a well-perfused organ; thus, the background signal from residual blood in the tissues appears minimal. 

## 4. Conclusions

In the present study, we showed that i.v. injected micelles containing multiple pendant aromatic groups (i.e., TMC-Bz monomers) in the hydrophobic blocks of the PCL-PEG-based block copolymers displayed longer circulation times in mice than micelles with a single terminal aromatic group, and that incorporating the pendant aromatic groups improved retention of the photosensitizer mTHPC in human plasma in vitro. Despite those promising features, similar biodistribution of micellar mTHPC as compared to free mTHPC—and importantly, unequal biodistribution patterns of mTHPC and the host micelles—indicated premature release of mTHPC from these micelles in vivo. Our study emphasizes the necessity to investigate the in vivo behavior, particularly biodistribution, of both the micellar carrier and the incorporated cargo, as such data can provide important information about the fate of PS-loaded nanocarriers. Our study shows that additional measures beyond π-π stacking are needed to stably incorporate mTHPC in the micelles in order to benefit from them as carriers that are able to deliver the payload in pathological tissue by passive or active targeting.

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
