# Peer review of "π-π-Stacked Poly(ε-caprolactone)-b-poly(ethylene glycol) Micelles Loaded with a Photosensitizer for Photodynamic Therapy"

_pharmaceutics, 2020, doi:10.3390/pharmaceutics12040338_

Round 1

Reviewer 1 Report

This study by Liu et al submitted to Pharmaceutics (Manuscript ID: pharmaceutics-758642) should be of interest for the readers of the journal. 

Only minor revision needed:

Line. 267-264: Full name of MTS + Where from? + Time of MTS after light exposure?

Line. 299: Specify that «cardiac puncture» were «conducted under deep terminal anaesthesia».

Line 313: «To 100 mg of sliced samples»?

Figure 7, 8 and 9. Why was the skin omitted? Very important tissue that should have been included, especially when this is in the field of PDT.It is written in the m&M (line 309) that it was included in the biodistribution study.

Line 728- 743: Conclusion should be shorter and more to the point.

Refs: Include doi:

Reviewer 2 Report

In this manuscript entitled, π-π stacked poly(ε-caprolactone)-b-poly(ethylene 2 glycol) micelles loaded with photosensitizer for 3 photodynamic therapy by Liu et al, the authors encapsulated the mTHPC into micelles modified by the introduction of aromatic rings by copolymerization of CL with TMC-Bz. Subsequently, the authors analyzed the in vitro stability, cytotoxic potential and in vivo redistribution of constructed micelles.

Following are the observations and specific comments to be considered for the improvement of the manuscript.  

1. In the Introduction section, the statements about PDT are too affirmative. PDT is still mostly used as a palliative therapy of cancer due to its limited effectiveness and relapses after the treatment. Moreover, an increased number of publications indicates the immunosuppressive outcome of the PDT. Therefore, the description of PDT needs further attention.

  2. The authors show no "dark cytotoxicity" of mTHPC loaded micelles. Nevertheless, it was also indicated that micelles, especially in FBS containing medium, are less effectively uptaken by the tumor cells [Kiesslich T et. al, Photochem Photobiol Sci. 2007 Jun;6(6):619-27] Is it possible that ineffective uptake of constructed mTHPC-micelles caused rather negative results in in vivo tests? How effectively the constructed micelles are uptaken by the cells?   3. Figure 6, especially 6B is illegible. It should be improved by reducing the points on the OX axis since there are no differences above 15 h post injections.   4. The mouse organs have been analyzed without perfusion. Therefore, the results are affected by mTHPC-micelles dissolved in the blood. It should be at least tested whether the perfusion of blood can change the results, especially when the liver, spleen, and kidney show the highest accumulation of photosensitizer.    5. The authors tested the accumulation of mTHPC-micelles, however, they did not perform the anty-tumor in vivo treatment. The reason is only the effectiveness of the uptake of mTHPC-micelles?

Round 2

Reviewer 2 Report

The authors improved the manuscript. In my opinion, no further corrections are needed.